# Exploring the Main Determinants of National Park Community Management: Evidence from Bibliometric Analysis

Yangyang Zhang [1,2], Ziyue Wang [1], Anil Shrestha [2], Xiang Zhou [1], Mingjun Teng [1], Pengcheng Wang [1,*] and Guangyu Wang [2,*]

1   College of Horticulture and Forestry Sciences, Hubei Engineering Technology Research Center for Forestry Information, Huazhong Agricultural University, No. 1 Shizishan Street, Hongshan District, Wuhan 430070, China; yangyangzhang1993@outlook.com (Y.Z.); yunjiuwang1008@gmail.com (Z.W.); zhouxiangw@webmail.hzau.edu.cn (X.Z.); tengmj@mail.hzau.edu.cn (M.T.)
2   National Park Research Centre, Faculty of Forestry, University of British Columbia, 2424 Main Mall, Vancouver, BC V6T 1Z4, Canada; anil.shrestha@ubc.ca
*   Correspondence: wangpc@mail.hzau.edu.cn (P.W.); guangyu.wang@ubc.ca (G.W.)

**Abstract:** The establishment of protected areas such as national parks (NPs) is a key policy in response to numerous challenges such as biodiversity loss, overexploitation of natural resources, climate change, and environmental education. Globally, the number and area of NPs have steadily increased over the years, although the management models of NPs vary across different countries and regions. However, the sustainability of NPs necessitates not only effective national policy systems but also the active involvement and support of the local community and indigenous people, presenting a complex, multifaceted challenge. Although the availability of literature on community-based conservation and NPs has increased over the years, there is a lack of research analyzing trends, existing and emerging research themes, and impacts. Hence, in this study, we employed bibliometric methods to conduct a quantitative review of the scientific literature concerning community management of NPs on a global scale. By analyzing data from published articles, we identified research hotspots and trends as well as the quantity, time, and country distribution of relevant research. We developed a framework to illustrate the main research hotspot relationships relevant to NPs and community management, then summarized these findings. Based on the literature from 1989 to 2022, utilizing 2156 research papers from the Web of Science Core Collection database as the data source, visualizations were conducted using the VOSviewer software (1.6.18). Based on the results of network co-occurrence analysis, the initial focus of this field was on aspects of resource conservation. However, with the convergence of interdisciplinary approaches, attention has gradually shifted towards human societal well-being, emphasizing the "social-ecological" system. Furthermore, the current research hotspots in this field mainly revolve around issues such as "natural resources, sustainable development, stakeholder involvement, community management, sustainable tourism, and residents' livelihoods". Effectively addressing the interplay of interests among these research hotspot issues has become an urgent topic for current and future research efforts. This exploration necessitates finding an appropriate balance between environmental conservation, economic development, and human welfare to promote the realization of long-term goals for sustainable development in NPs.

**Keywords:** national parks; community management; bibliometric analysis; sustainable development

## 1. Introduction

Natural protected areas (NPAs) are widely recognized as crucial elements for biodiversity conservation, natural resource management, and sustainable development. They play a vital role in safeguarding global species and ecosystems. However, the tension between conservation and development is a long-standing and globally prevalent issue [1–3]. As NPAs rapidly expand, the interactions between protected sites and local communities grow

increasingly complex [4]. Paying attention to the interests of local community residents and allowing the local people and indigenous community to participate fairly in the management of NPAs will benefit the long-term conservation strategies of NPAs "Protected Planet Report, 2020" [5]. Moreover, aligning with the "2030 Agenda for Sustainable Development", which emphasizes human well-being and quality of life, is necessary to maximize benefits and minimize losses; these considerations significantly impact social support and natural conservation of NPAs [6]. Based on the perspective of the interrelationship between humans and nature, scholars and government policymakers must recognize the profound importance of the bidirectional and dynamic relationship between humans and nature for a sustainable and resilient future.

The International Union for Conservation of Nature (IUCN) has classified protected areas into six categories, with national parks (NPs) falling into Category II [7]. Since NPs are not as strictly protected as Category Ia (strict nature reserve), with more stringent protection measures, or Category Ib (wilderness area), with limited visitor access, NPs are designed to achieve balanced development of ecological, economic, and social benefits both within and surrounding the park [8]. This objective is realized through various approaches, such as community-based conservation, co-management, and integrated conservation and development [9]. NPs represent a distinctive "socio-ecological" complex ecosystem that fosters an intricate relationship between nature and the community nexus [10,11]. Gaining a profound understanding and clear delineation of this relationship is critical for fostering harmonious coexistence between humans and nature while safeguarding biodiversity.

In the context of setting primary objectives for NPs, there is a strong emphasis on the significance of preserving biodiversity and natural processes. Simultaneously, due consideration is given to the social needs of indigenous inhabitants striving to achieve sustainable development goals within local communities [7]. Consequently, the role of communities in global conservation is being increasingly prioritized. The establishment of NPs has significant social, political, institutional, economic, and environmental impacts on surrounding communities [12]. Since the inception of the world's first NP, Yellowstone NP in the USA in 1872, numerous countries and regions, including Canada, Australia, New Zealand, South Africa, and others have followed the Yellowstone model and established their own NP systems [13]. Indeed, the prevailing model of NPs has historically treated human settlements and livelihood activities as incongruent with the conservation and recreational objectives of the parks. Consequently, indigenous inhabitants living within designated NP areas were often evicted and resettled to create a "wilderness" landscape, devoid of human habitation, to prioritize nature preservation [14–16]. However, this approach disregards the rights of indigenous peoples, limiting their access to resources and even depriving them of the right to manage or utilize NP resources. The social consequences of such strict control over resources are profound; displacement, economic hardships, altered livelihoods, increased poverty, and intergroup conflicts often result [17,18]. Such policies have also resulted in unequal allocation of public resources [19], ultimately hindering the achievement of the conservation objectives originally planned for NPs.

In the 1980s, there was a general concern about poverty and vulnerability around NPs. Additionally, the realization that exclusionary approaches could backfire prompted managers to incorporate the needs of local communities into NP management [20]. Inevitably, community residents need to directly or indirectly utilize the natural and biological resources within NPs to sustain their livelihoods [21–23]. This includes engaging in traditional practices such as timber harvesting, medicinal plant collection, hunting, fuelwood collection, and grazing. With the adjustment of the socio-economic structure, community residents have reduced their demand for traditional natural resources and become more actively involved in the tourism service industry. NP managers have also proactively developed community-based tourism services [20]. On one hand, community residents promote unique local products such as coffee, tea, honey, and traditional medicinal herbs to tourists, and provide additional services like accommodations, dining, and hospitality. On the other hand, this community tourism development model generates a substantial

number of employment opportunities for the local community, improving their livelihoods and contributing to social, cultural, and ecological conservation [24]. This ensures the sustainable development of NPs. As a result, the relationship between NPs and local communities is inseparable.

Vulnerabilities faced by local communities, such as poverty, food insecurity, and conflicts over resource use, can jeopardize biodiversity conservation efforts in NPs [25]. It is difficult to achieve the management objectives of NPs without the support and involvement of local people [17]. The IUCN emphasizes that conservation is not the sole objective of NPs—conservation policies must work in tandem with indigenous ownership and management rights over NPAs [16,18]. Conservation policies now aim to balance ecological conservation and social development, acknowledging that promoting the well-being of local communities and indigenous peoples is of equal importance.

Bibliometric reviews are often used to illuminate general patterns in academic literature [26]. Bibliometric or quantitative analyses of scientific publications are commonly used to examine the development trends in various research fields [27,28]. In this paper, by comprehensively collating and summarizing the bibliometric data on the current research status, research hotspots, and regional distribution, we visually present the development trends in NP community management research. Based on this analysis, we elaborate on the characteristics and correlations of recent hotspot issues faced by NP community management in recent years.

## 2. Materials and Methods

### 2.1. Data Collection

To ensure the comprehensiveness of our research, we considered the diverse development courses of NP communities in various countries. To encompass a wide range of relevant words within the scope of our research topics, we conducted a focus group to identify synonyms or near-synonyms of the term "Community". The aim was to include as many relevant terms as possible in our analysis.

Using the Web of Science (WOS) core collection database, we searched for titles, abstracts, author keywords, and Keywords Plus with the logical relationship "intersection" (AND). The keyword strings used were TS = ("National Park*") AND TS = ("Community Management*" OR "Community-Based Management*" OR "Community Conservation*" OR "Local People*" OR "Buffer Zone*" OR Co-management* OR Indigenous*). A total of 2156 unique articles were obtained. We found that the earliest paper appeared in 1989, so the period was from 1 January 1989, to 31 December 2022.

We chose to extract these document types because they represent peer-reviewed certified knowledge. Using the "Export Records to Tab Delimited File" option in WOS, we selected "Full Records and Cited References" as the Record Content. The metadata of the WOS database research was then exported as a ".text" file and imported into VOSviewer (Version 1.6.18) [29] for detailed analysis.

### 2.2. Data Processing

VOSviewer (https://www.vosviewer.com/ (accessed on 1 May 2023)) is a free bibliometric analysis and visualization software [30] that excels in "co-occurrence" network clustering as well as density analysis, generating network graphs that position displayed items based on their degree of association. Strongly correlated items are placed close to each other, while weakly correlated items are farther apart. To explore major topics relevant to NP communities, we generated a co-occurrence map of themes. Additionally, we used Microsoft Excel 2016 for a descriptive analysis of the number of articles, primary sources of publications, and countries represented in the database [26].

To visualize the most important keywords, we selected the frequency of title and abstract fields in VOSviewer's "Choose fields" option, then selected "Ignore structured abstract labels" and "Ignore copyright statements". To avoid redundancy in the generated network diagram and ensure an accurate representation of the results, we used a synonym

file to unify keywords involving the same topic (replace or merge) and re-imported the new file for operation. VOSviewer's overlay visualization application allows network items to be represented on a temporal gradient, representing the co-occurrence of network items as a function of time. This visualization is based on the average publication year of documents in which keywords appear [31]. For enhanced visualization, we set a criterion in VOSviewer by which words had to appear more than 30 times, resulting in 57 words. After replacing or merging synonyms and ignoring invalid words, we obtained a final set of 45 high-frequency keywords.

## 3. Results

### 3.1. Visual Presentation of Research Topic

In the co-occurrence map, the 45 resulting topic keywords were grouped into three distinct clusters (Cluster 1: 22; Cluster 2: 14; Cluster 3: 9) (Table 1). The size of each keyword in the map is proportional to its frequency of co-occurrence (Figure 1).

**Table 1.** Clusters and relative keywords resulting from the co-occurrence analysis of keywords.

| Cluster 1 (Red) | | | Cluster 2 (Green) | | | Cluster 3 (Blue) | | |
|---|---|---|---|---|---|---|---|---|
| K | O | T | K | O | T | K | O | T |
| National-Park | 514 | 1474 | Management | 336 | 1039 | Protected Areas | 194 | 683 |
| Conservation | 505 | 1347 | Biodiversity | 267 | 868 | Local People | 185 | 708 |
| Forest | 180 | 492 | Policy | 80 | 215 | Wildlife | 130 | 502 |
| Community | 151 | 479 | Africa | 70 | 200 | Attitudes | 122 | 495 |
| Impact | 142 | 396 | Tourism | 61 | 181 | Perceptions | 106 | 439 |
| Patterns | 119 | 341 | Governance | 60 | 181 | Reserve | 84 | 357 |
| Diversity | 97 | 227 | Participation | 49 | 173 | Conflict | 77 | 279 |
| Population | 77 | 152 | Ecosystem Services | 42 | 147 | Livestock | 33 | 119 |
| Ecology | 64 | 138 | Land | 42 | 128 | Behavior | 31 | 73 |
| Knowledge | 64 | 156 | Ecotourism | 39 | 165 | | | |
| Vegetation | 64 | 119 | Co-management | 36 | 103 | | | |
| Climate-Change | 55 | 127 | Poverty | 33 | 127 | | | |
| Land-Use | 54 | 159 | Resources | 33 | 104 | | | |
| Dynamics | 51 | 105 | Livelihoods | 31 | 131 | | | |
| Deforestation | 47 | 133 | | | | | | |
| Sustainability | 46 | 142 | | | | | | |
| Landscape | 43 | 118 | | | | | | |
| Buffer Zone | 37 | 99 | | | | | | |
| Abundance | 33 | 72 | | | | | | |
| Medicinal-Plants | 32 | 71 | | | | | | |
| Areas | 31 | 78 | | | | | | |
| Region | 31 | 68 | | | | | | |

K: keywords; O: co-occurrences (frequency of keywords); T: total link strength (the cumulative strength of the links of an item with other items).

Cluster 1 includes the core keywords "National-Park" and "Community", which serve as the foundation for exploring research hotspots between these two topics. This cluster includes themes such as "Conservation", "Forest", "Diversity", "Ecology", "Ecology", "Climate-Change", "Land-Use", "Deforestation", and "Sustainability". Each of these words co-occurred more than 46 times, with their total co-occurrence with other keywords exceeding 142 times, indicating significant correlation and interconnectedness among them.

Cluster 2 centers around the core keywords "Management" and "Co-management", encompassing keywords related to participation in NP management, such as "Policy", "Governance", and "Participation". Their co-occurrences for these terms were 80, 60, and 49, respectively. This cluster also involves "Biodiversity", "Ecosystem Service", and "Livelihoods". Among them, "Biodiversity" had the highest co-occurrence (267) while "Livelihoods" had the lowest (31).

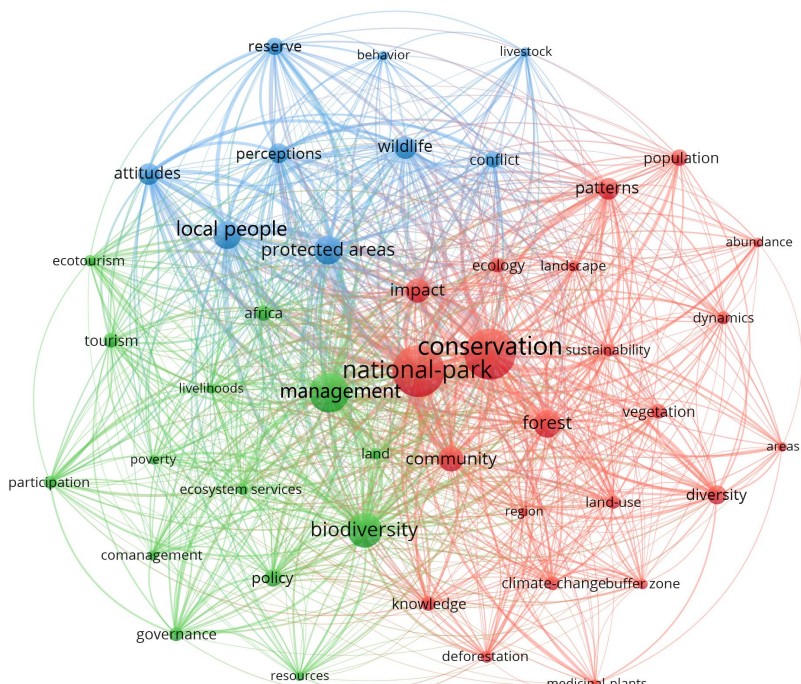

**Figure 1.** Co-occurrence network map of keywords in the global scientific literature on NPs and community management. The size of each keyword (node) in the network is directly proportional to its number of occurrences in the documents analyzed. Colors indicate clusters to which keywords are univocally assigned based on their reciprocal relatedness. Circle size is weighted by the number of occurrences. Line width indicates the strength of the symbiotic relationship.

Cluster 3 contains the fewest words and encompasses terms related to local attitudes towards protection such as "Attitudes", "Perceptions", "Conflict", and "Behavior", as well as terms related to "Wildlife" and "Livestock". Among them, "Attitudes" and "Perceptions" appear 122 and 106 times, respectively; "Behavior" has the lowest frequency of occurrence at 31 instances.

Visually (Figure 1), Cluster 1 (in red) shows a high degree of overlap with Cluster 2 (in green), as some keywords are positioned between the network regions of the two clusters. Notably, "National Park" and "Conservation" from Cluster 1 are closely related to "Management" from Cluster 2, with a highly interconnected intersection area. These terms also have the largest circle areas, reflecting the prioritization of conservation as the primary management goal in NPs. In contrast, Cluster 3 (in blue) is located at the top of the entire network graph, with less overlap and occupying a smaller area relative to Clusters 1 and 2.

Overall, the entire left part of the network diagram in Figure 1 illustrates a relationship between NPs and human welfare, along with issues related to participation in management policies. This includes various aspects such as "Biodiversity", "Ecosystem Services", "Land", "Resources", "Livelihoods", "Participation", and "Policy". On the right part of the diagram, the research focuses on the sustainability of the ecological environment in NPs, incorporating topics such as "Diversity", "Climate Change", "Land Use", "Deforestation", "Wildlife", "Conflict", and "Sustainability".

Considering the complexity of social-ecological systems, there can indeed be potential conflicts between the sustainable protection and development of NPs and the social welfare of indigenous communities. An effective approach to addressing this challenge is to adopt a co-management model based on community participation. By enhancing the policy system of NPs, the actual development needs of the community can be incorporated into the planning and management of NPs. This would be a crucial step towards achieving sustainable development.

Figure 2 shows an overlay visualization map based on the publication years of the documents, which illustrates the evolution of scientific research on NPs and community management. The color of the nodes indicates the average year for each keyword, pinpointing the most recent themes and research trajectories [27,31]. The overlaid visualization demonstrates that recent attention has been directed toward human-oriented research methods, policies, and areas of interest in the context of community engagement of NPs. Keywords such as "Participation", "Governance", "Ecosystem Services", "Medicinal Plants", "Livelihoods", and "Climate Change" appear to have garnered increased attention in recent years.

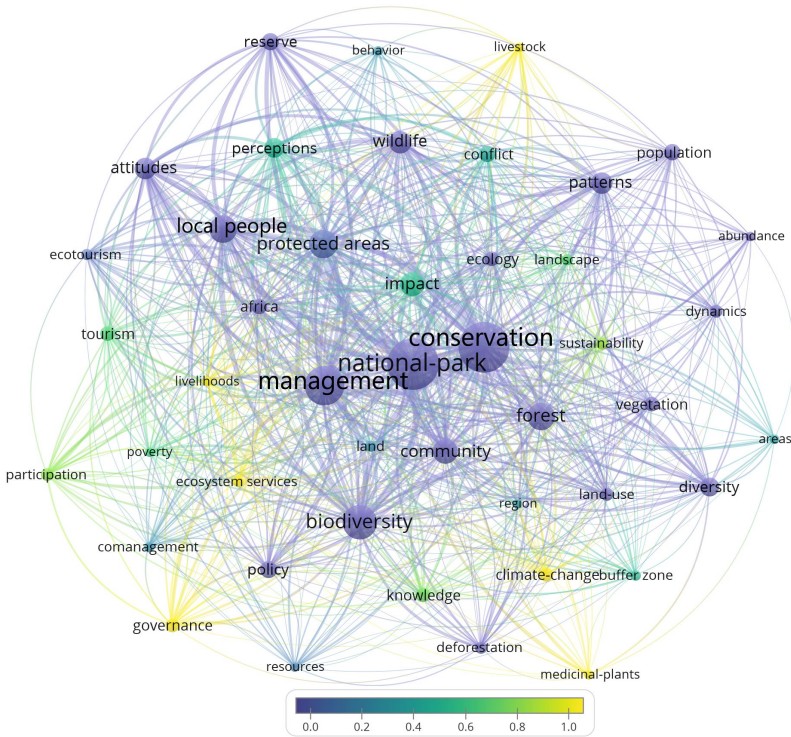

**Figure 2.** Overlay visualization of the co-occurrence network map of keywords. Keywords are represented based on the average year of publications of documents they occur in, on a color gradient from blue (older publications), to purple (publications equally distributed across the timespan 1989–2022), to yellow (more recent publications).

Other keywords, including "National Park", "Management", "Community", "Conservation", "Biodiversity", "Forest", and "Local People", are evenly distributed within the 0–1 range corresponding to publication years from 1989 to 2022. This indicates that these topics have maintained enduring relevance and consistent appearances in the literature on NPs and community management throughout the history of this research field. Judging from the score index, research on NP communities has transitioned from focusing solely on natural resource protection to placing greater emphasis on human social welfare since the year 2009 (score index of 0.6).

### 3.2. Visualization of Number of Published Papers and the Global Distribution of Literature

We established the visualizations shown in Figures 3 and 4 by searching the 2156 articles in the core collection of WOS and analyzing publication years and author nationalities. Based on Figure 3, we can divide the development of relevant publications into three stages. Firstly, from the earliest related publication in 1989 to before 2000, there were at most 24 publications per year, indicating a slow development phase (Phase I). Secondly, from 2001 to 2017, the number of publications steadily increased, representing a steady development phase (Phase II), with an annual average of no more than 100 articles. Thirdly, after

2017, the number of publications entered a rapid growth phase (Phase III), reaching its peak in 2020 (209 articles) and slightly declining thereafter.

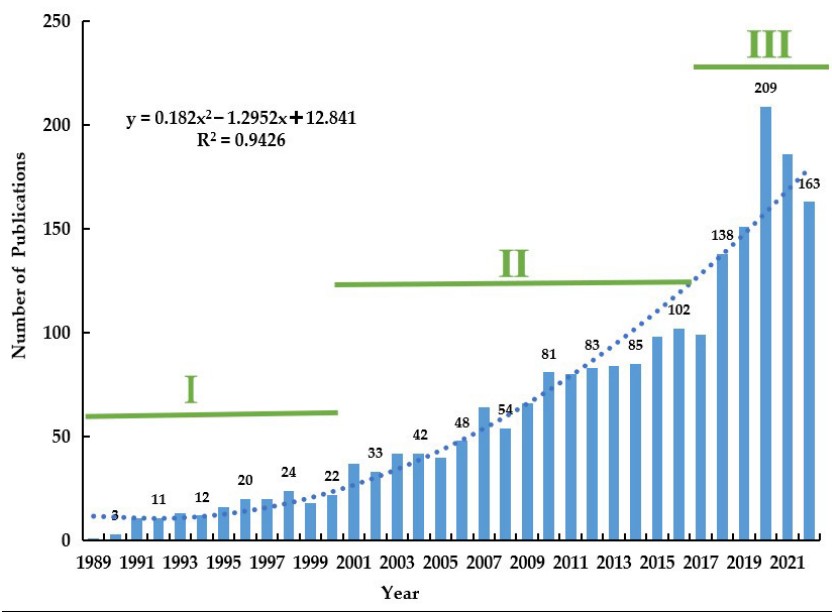

**Figure 3.** Number of articles published by NP community management from 1989 to 2022. Phase I, slow development phase; Phase II, steady development phase; Phase III, rapid growth phase.

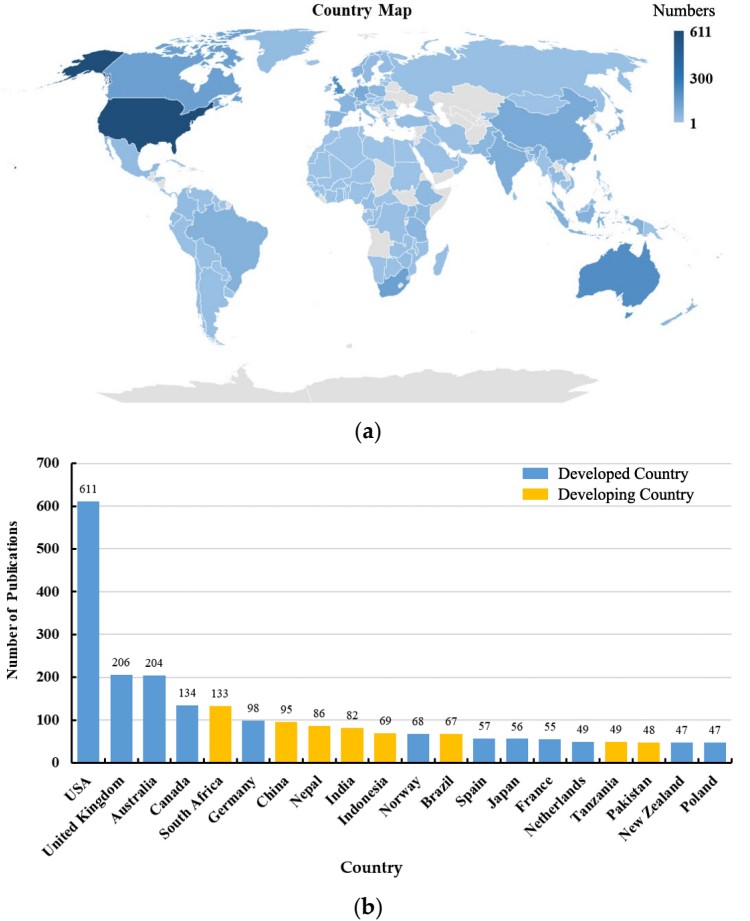

**Figure 4.** (**a**) World map illustrating the distribution of publications related to NP community management from 1989 to 2022; (**b**) top 20 countries ranked by the total number of publications.

Additionally, the bar graph of annual publication volume shows a high fitting coefficient, $R^2 > 0.9$, indicating a favorable predictive effect. The trend line further forecasts that by 2030 (corresponding to the "United Nations Sustainable Development Agenda 2030"), the annual research publication volume on this topic could reach approximately 266 articles.

Figure 4a,b shows the geographical distribution of articles related to community management in NPs worldwide and presents the top 20 countries or regions of the authors contributing to these publications. The varying shades of blue represent the number of articles published, with darker colors indicating a higher number. The grey areas represent regions where articles related to this topic have not yet been published in the WOS (Figure 4a). Notably, several regions in Africa and Asia are depicted in grey, suggesting a limited publication of articles on this topic or minimal collaboration with other countries. Examples of such countries in Asia include North Korea, Laos, Kazakhstan, Uzbekistan, Turkmenistan, Afghanistan, Tajikistan, Yemen, Syria, Lebanon, and Georgia. In Europe, they include Belarus, Ukraine, Moldova, and Bulgaria; in Africa, they include Liberia, Angola, South Sudan, the Republic of Chad, and Somalia. Central American countries such as Guatemala and Nicaragua are also illustrated in grey. A common characteristic among these areas is that they are predominantly developing countries.

On the other hand, countries or regions that established NPs early on, such as the United States (Yellowstone NP, 1872), Australia (Royal NP, 1879), Canada (Banff NP, 1885), and the United Kingdom (Peak District NP, 1932), have a substantial body of existing literature focusing on the management issues of NPs. The abundance of research findings in these countries is also influenced by the timing and number of NPs established, reflecting the historical development and evolution of NPs and their management practices.

Figure 4b shows the top 20 countries in terms of the total number of publications related to NP community research spanning the past 33 years. Leading this list is the USA, with a total of 611 publications, followed by the United Kingdom (206) and Australia (204). Upon examining the level of national development, developed countries were found to account for 3/5 of the top 20, while developing countries (South Africa, China, Nepal, India, Indonesia, Brazil, Tanzania, and Pakistan) account for 2/5. These countries are predominantly situated in Asian or African regions, with South Africa having the largest number of publications, followed by China and Nepal. The figure also shows that the total number of papers published in developing countries (629) is substantially lower than that in developed countries (1632), accounting for only about 27.82% of the total number of publications.

In summary, our analysis indicated a steady increase in the number of articles published in the field of global NP community management, signifying growing interest among scholars. However, it is important to note that a significant majority of these articles were authored by researchers from developed countries, which have a higher level of investment and output in NP research. Research in this field remains relatively limited in developing countries. This disparity may stem from various factors that warrant further investigation.

## 4. Discussion

Based on the findings shown in Figures 1–4 and a comprehensive review of the relevant literature, we have conducted a specific analysis of the community management of NPs. Throughout more than 150 years of development, NPs have been associated with various pressures and challenges arising from human activities and changes in the climate and natural environment. In response, efforts have been made to encourage public participation in NP management through information sharing and educational initiatives, to delineate protected area boundaries, and to prioritize improved relationships with local indigenous communities [32,33].

However, to successfully achieve these goals, several issues must be addressed, including but not limited to the following prominent research topics: NP management systems [4,34], conflicts of natural resource use, community participation [13,35], com-

munity livelihood issues [36–38], human–wildlife conflicts [39], and the protection of ecosystems [40] and biodiversity [41]. A comprehensive understanding of the relationships between these issues is imperative. We developed a relationship framework encompassing the main research hotspots in NPs as shown in Figure 5.

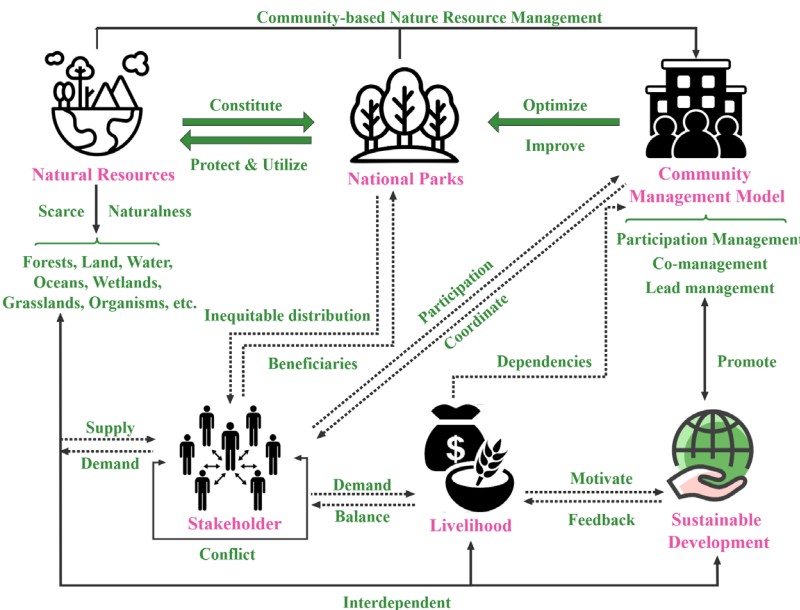

**Figure 5.** Relationship framework among the main research hotspots in NPs.

### 4.1. Complex Relationship between National Park Communities and Natural Resources

Since the early 1990s, strategies such as "Community-based Natural Resource Management" and "Community-based Conservation" have emerged, aiming to involve and empower local communities as significant participants and decision makers in natural resource management. It is essential to understand that natural resource management operates within a broader socio-ecological context, requiring decisions to be made with consideration of both environmental and societal impacts including those related to protected areas [42]. This management approach, when applied to NPs or other NPAs, pursues a dual objective: the conservation of natural resources and sustainable development. It also aims to strike a balance between the interests of local communities and conservation organizations [43].

Table 1 shows our integration of three sets of keywords related to natural resource management, including "Forest", "Climate Change", "Land Use", "Sustainability", "Biodiversity", "Ecosystem Services", and others. These keywords reflect the interconnectedness within the research domain of this study. Moreover, they encompass various policy management patterns, including those related to "Patterns", "Policy", "Governance", "Participation", and "Co-management". Given the existing imbalance between the actual needs of local community residents and the supply of natural resources within NPs, stakeholders often hold distinct attitudes and viewpoints regarding the conservation and management of these areas. These differences inevitably lead to conflicts (Table 1, Cluster 3, keyword "Conflict").

The primary challenge in balancing conservation and development lies in effectively maintaining and utilizing natural resources. Striking a balance between these two aspects is a key issue. NPs and their surrounding areas possess a wealth of natural resources, including but not limited to forests, land, oceans, wetlands, glaciers, grasslands, and wildlife, as well as spectacular scenery, geological features, and cultural diversity [44]. These natural resources are not only crucial for the survival and livelihood of community residents but also play a significant role in promoting economic development [45,46]. On

one hand, natural resources are a source of income and a means of poverty alleviation for marginalized communities. On the other hand, they are a major trigger for conflicts over their usage [47].

The use of natural resources is intricately linked to the development model of NP communities. One example is the management of land resources (Table 1, keywords "Land", "Land Use"), where the exclusive land management model traces back to 1000 BC and was later formalized through the creation of the special protection mechanism of NPs. This concept was first proposed in the USA and has since been adopted globally. However, this model has significant drawbacks. It infringes on the rights and interests of indigenous people, as they are often excluded from these areas [48]. Local communities that traditionally depended on forest resources and other ecosystem services for their livelihoods have been restricted from accessing them after the establishment of NPAs [49]. Further, this model can lead to poverty and social problems in local communities [18], where residents must heavily rely on the resources within NPAs to sustain their livelihoods. This dependence, in turn, threatens the original intention of protecting biodiversity [25]. The establishment of Yellowstone NP, the world's first NP, is an example of the consequences of this wilderness protection ideology. It shielded the habitats within its boundaries from external land use changes [50] at a cost to the indigenous inhabitants, such as Shawnee Indians, who were deliberately removed from their ancestral lands as up to 300 of them were killed in the name of nature protection [18,51].

Numerous studies have shown that the legal recognition of indigenous collective land ownership is highly conducive to the long-term development of NPs. For example, exceptional biodiversity conservation results have been observed in regions such as Canada, India, Bolivia, Brazil, and the Amazon basin of Colombia [52–54]. In contrast, developing countries face distinctive challenges in the management of NPs or NPAs, with escalating disputes over land use and significant collective land disputes being prevalent issues [17,55,56]. These situations have generated dissatisfaction and tension among local community members towards NPAs [36]. Therefore, a global phenomenon of "human-land" conflicts has emerged in NPs, underscoring the need for effective and equitable land-use planning and management policies to address and resolve land-related issues.

*4.2. Sustainable Conservation and Development of National Parks*

The development strategy for most NPs revolves around ecological conservation, natural resource preservation, and controlled tourism development to ensure sustained protection within limited areas. The data visualizations shown in Figures 1 and 2 highlight the prominence of the keyword "Conservation" among the 45 keywords we analyzed (apart from "National Park" and "Management"). Linked terms such as "Forest", "Deforestation", "Sustainability", "Ecology", "Vegetation", "Biodiversity", "Ecosystem Services", and "Ecotourism" all reflect the emphasis on sustainable ecological protection in NP management.

The sustainable development of NPs is closely related to community management and is influenced by the economic context of the respective country or region. As scholars deepen their research on the sustainable development and environmental protection of NPs, both developed countries such as the USA, Australia, the UK, and Canada, as well as developing countries, such as South Africa [24], China [57], Nepal [58], and India [17], have increasingly shown interest in these issues. It is noteworthy that a substantial number of natural parks, exceeding 5000, have been established across more than 200 countries and regions worldwide. These natural parks encounter various challenges pertaining to the sustainability of community resource management [8]. Nevertheless, defining precise functional zones within NPs can effectively safeguard ecosystem integrity and legality in the area, promote the rational utilization of natural resources, and facilitate community integration through feasible management strategies [59–61]. Such practices can effectively alleviate the pressures posed by these challenges and foster the sustainable development of NPs.

In the sustainable management process of NPs, resolving potential conflicts between ecological conservation and regional socioeconomic development, as well as between regional ecological benefits and the interests of local community residents, is a critical issue [62,63]. Socioeconomic and cultural factors significantly influence conservation decisions in most developing countries [64]. In these countries, communities adjacent to NPAs often face poverty as a primary socio-economic issue [65]. Striking a balance between the original goal of NPs, which is to protect and maintain natural resources and ecosystems, while also considering local economic development, is particularly challenging. Large NPAs that extend across national borders, cover multiple political jurisdictions, and include large indigenous populations present significant challenges [66,67].

Exploring win-win solutions for ecological conservation and community development, as well as understanding the relationship between ecosystem services and community well-being [68], lies at the core of effective and sustainable NP or NPA management. This calls for integrated and scientific practices, with a rational implementation of sustainable objectives.

### 4.3. Balancing Stakeholder Interests

According to stakeholder theory, any group or individual who has influence or is influenced by a set of planning goals should be considered as a stakeholder [69]. In the case of NPs, the population within and around the parks can be substantial—up to tens of millions of people. These individuals and groups, including government departments, community residents, franchisees, visitors, and others, all have a stake in the management and protection of NPs [70]. As stakeholders, they often have diverse interests that may lead to conflicts and competition; the prevailing trend, however, is toward coordination and cooperation [27,71].

Conflicts among stakeholders present serious challenges to the management of NPs and other NPAs. As shown in Figure 1, the core keywords of Cluster 3, such as "Protected Areas", are associated with "Local People", "Attitudes", "Conflict", "Perceptions", and "Management" in Cluster 2. This indicates that the effectiveness of managing NPs and NPAs is closely linked to the attitudes of community residents and conflicts arising among stakeholders. Therefore, for the sustainable development and management of tourism in NPs and other NPAs, managers need to understand stakeholders' perspectives (keywords "Attitudes", "Perceptions", Cluster 3) towards conflicts, challenges, and opportunities [72]. Stakeholders' positive attitudes towards NPs and other NPAs are not only the foundation for effective community action but also a key factor determining the success of sustainable development plans [73,74], which will help protect local ecosystems and biodiversity [75].

Additionally, management departments should adopt an open perspective when considering stakeholders and view them as partners rather than opponents. Implementing positive and humane policies to gain community residents' support for protection policies will facilitate the implementation of long-term protection strategies for NPs or NPAs, as highlighted in the "Protected Planet Report, 2020" [5]. However, it is essential to recognize that fully resolving the contradictions between NPs or NPAs and community interests remains challenging within today's rapidly changing and uncertain environment [76]. Effective solutions require continuous efforts to foster positive relationships and collaborative approaches to conservation, considering the perspectives and needs of all stakeholders involved.

### 4.4. Community Management Models

With the continuous improvement of the NP system, recognition of the community's rights to participate in decision-making, planning, and benefit-sharing is growing. This not only enhances the understanding of community residents regarding the relationship between the use of NP resources and the integrity of the ecosystem but also positively impacts the sustainability of NP management. Community participation has become a core force in promoting protective measures in many NPs and their management agencies [13].

Without community participation, achieving sustainable NP management becomes quite challenging [35]. To reduce or avoid conflicts among stakeholders, the government may actively incorporate strategies of community participation in conservation into the decision-making process [77].

The core keyword "Management" in Cluster 2 (Figure 1) is closely related to the keywords "Policy", "Governance", "Participation", "Co-management", and "Resources", as well as "Community" and "Impact" in Cluster 1. This indicates that NP community management strategies need to include a variety of factors to incorporate the interests of the community, ultimately achieving sustainable community development in NPs. Currently, a new approach classifies community participation in conservation into three categories: "Community-involved Management", "Community Co-management", and "Community-led Management" [78]. These forms represent a continuum, with conservation being viewed more as a means rather than an end, granting the community the right to manage its resources. While these three forms complement and relate to each other to some extent, they differ in their emphasis and operational mechanisms.

First, "Community-involved Management" is the most basic management model and a core concept in NP management, planning, and policy decision-making processes. It is positively correlated with managers' perceptions of the success of NPs [34]. This model not only benefits the ecological protection of NPs but also the interests of community residents, making it a key issue in NP development [8]. It is also an important means for co-development between NPs and communities. Under this model, community members can suggest specific projects or decisions but do not necessarily have decision-making power. This approach helps maintain enthusiasm among community residents, improves NP management efficiency [79], and enhances the legitimacy and rationality of protection decisions. However, achieving genuine community participation often faces many challenges in practice. In most cases, residents participate passively in the management of NPs, and they are often only informed about decisions that have already been made or events that have already occurred [80]. Various research results indicate that the definition, degree, and scope of community participation may be influenced by an array of factors such as social, political, cultural, technological, and economic conditions [74,81–83]. Changes in these factors can lead to significant differences in the forms and effects of community participation [84].

Second, "Community Co-management", also known as collaborative management or joint management, is a more equal management model. Under this model, local resource users and other stakeholders formally share responsibilities, rights, and benefits in the co-management of NPs [85]. It is an inclusive strategy, originating from national and international policy settings, to promote the participation of all relevant actors [86]. However, the degree of community residents' participation in NPs, even as "gatekeepers", can be significantly affected by economic conditions. This is especially true in communities in economically underdeveloped areas [24].

Third, "Community-led Management" is defined by the community taking the leading role in the management process and can be viewed as a "bottom-up" management model. Under this model, decision-making and management activities are mainly carried out by stakeholder groups and the community or organizations representing the community. The NP community has significant influence over the management process and may even have the ultimate decision-making power [47]. The core concept of this model is that those who are closest to and most dependent on the park's resources should have the most say in the management and use of these resources.

The main differences between these three models lie in the degree of community participation and the amount of decision-making power in the management process. Regardless, they all emphasize the importance of community participation in nature conservation. Whether through direct or indirect participation, community involvement can help in identifying and resolving potential social problems and conflicts, prevent clashes between community residents and biodiversity conservation goals [87], and make policies more

targeted [55,88]. From the conservationist's perspective, the future development of many NPAs will be limited without the active participation and support of local communities. In any country or region, if there is a lack of active community participation, the management of NPs will fail to reach maximum effectiveness [16].

### 4.5. Tourism Development

Community residents should not only obtain economic benefits by actively participating in tourism, but they should also play a role in decision-making and management to support the development of sustainable tourism [24,25]. Sustainable tourism can stimulate ecological restoration, provide additional income to resolve resource-use conflicts, increase the income of community residents, reduce NP management costs, and gain broader community support. This has profound significance for NPs [89].

The core keywords "Tourism" and "Ecotourism" in Cluster 2 (Figure 1) are closely associated with the keywords "Livelihoods", "Management", and "Poverty", revealing a relationship between tourism development and NP management. The tourism industry is crucial in guiding NP communities to participate in and support ecological conservation [90]. The key to community-based tourism development lies in effectively managing NPs to ultimately achieve regional sustainable development. Sustainable tourism centered on NPs can provide economic as well as environmental benefits to the community.

NPs can attract tourists from all over the world [91], bringing new economic resources to local community residents [25]. This contributes to social, cultural, and ecological conservation and promotes the development of related industries [55]. It also prevents economic activities that cause environmental damage, eliminating concentrated protective development traps [92]. This provides multiple job opportunities for community residents, effectively solving livelihood problems otherwise caused by strict NP policies [16]. As important venues for eco-tourism, NPs not only provide entertainment and recreation opportunities to the public but also are widely recognized by scholars and practitioners as important contributors to protection and development. However, achieving the dual goals of nature protection and sustainable development requires effective stakeholder cooperation [93]. This includes conserving natural ecosystems and improving the livelihoods of local communities through eco-tourism strategies [94]. By striking a balance between these goals, NPs can maximize the positive impact on both the natural environment and the well-being of the local population [95].

Simultaneously, it should not be overlooked that the vigorous growth of the tourism industry has brought about certain challenges for the sustainable development of NPs. The substantial tourism activities around NPs unavoidably inflict a certain degree of damage and disruption upon the natural systems, concurrently leading to shifts in the socio-cultural environment [15]. Nevertheless, by actively guiding community engagement and implementing effective management measures, NP managers can minimize these disturbances, thereby achieving the dual objectives of ecological conservation and sustainable development for the NP [96,97].

### 4.6. Livelihood Concerns for Residents of National Park Communities

The livelihoods of local community members are deeply influenced by the policies and management styles of NPs, and the impact of NPs on local livelihoods may be the main determining factor of the attitudes towards conservation among residents [36,98]. The relationship between NPs and natural resource management is shaped by two important factors: ecosystem services and community livelihoods [11]. This is evident in the core keywords in Cluster 2 (Figure 1): "Livelihoods", "Management", "Ecosystem Services", "Policy", and "Resources". As conflicts arise between the demands for improving community livelihoods and the goals of protecting biodiversity, policy interventions often exacerbate the socio-ecological dilemmas faced by residents [99].

The conservation strategies of NPs create livelihood-related pressure on the surrounding community, especially when it is already suffering from impoverished conditions. NPs

are usually located in relatively remote and often marginalized rural areas, where the rural economy is typically underdeveloped [100,101]. The livelihoods of rural residents mainly rely on natural resources; when the use of such resources is limited, many families may see their livelihoods threatened [47,102]. The reasons for this are mainly related to the development model of NPs.

Private production and lifestyle-related activities are prohibited within the core areas of NPs, which creates restrictions on the traditional livelihoods of indigenous communities. Traditional practices such as agriculture, forestry, and livestock farming may be strictly limited after the establishment of an NP [16,103–105]. NP managers should consider optimizing the structure of community livelihood resources, increasing economic income to improve community infrastructure, providing job opportunities, and ultimately promoting the overall development of NP communities to offset their negative effects on indigenous peoples [8,106].

The resource utilization methods and conservation strategies of NP communities play a decisive role in their relationship to natural resources [26]. In some cases, limited resource access may force residents to resort to illegal practices, such as mining [107], poaching, agricultural encroachment, fishing, hunting, logging, and gathering, to sustain their livelihoods [16,85,101]. Furthermore, urbanization and agricultural expansion are considered the two major driving factors leading to deforestation. The logging of forests results in a gradual degradation of forest services and has both direct and indirect impacts on rural and urban societies [108]. The formulation of national policies should involve a comprehensive understanding of the actual needs of community residents. However, the reality is that this understanding often neglects their rights to oversee, manage, and utilize resources, while also suffering from a lack of necessary trust [109,110]. This triggers dissatisfaction and resistance from community residents towards NPs, leading to resentment against conservation policies [111]. In some areas, this may even lead to violent conflicts between community residents and the park rangers or guards of NPs [32,101,104]. Therefore, community residents are more likely to hold positive attitudes towards these policies when the benefits of NP-related policies outweigh the costs [91,112].

### 4.7. Environmental Issues Challenges Faced by National Park Communities

The environmental challenges faced by NP communities encompass various aspects, including ecological balance and conservation, protection of wildlife and plants, land development and utilization, water resource management, the impacts of climate change, sustainable tourism and visitor management, as well as community engagement and awareness enhancement. Taking natural resources as an example, communities that heavily rely on natural resources may hinder NP development, trigger environmental degradation, and even lead to disorder in the local social order. There is a clear contradiction between resource dependence and the development of rural areas near NPs [26]. Strict conservation policies and regulations of NPs have not only altered the local social structure and environmental patterns [27,37,81,113], but have also affected the income distribution among community residents, further intensifying competition for limited natural resources. To address these challenges, governments worldwide are establishing new "people-centered" NPAs, empowering grassroots institutions with legal powers to manage local resources. The aim is to change community residents' attitudes towards NPs [47,114] and reduce tensions between them and the NP.

In some populous countries and regions with high per-capita use of natural resource usage, top-down mandatory management strategies have been implemented by the government which, to some extent, disrupt the sustainability of NPAs [76,115]. While such strategies may achieve the "Unified, Standardized, Efficient" management objectives of NPs [116], they also come with drawbacks. Under the guise of protection, these strategies may overlook the rights of local communities [117], resulting in limited or no participation of community residents in NP management decisions [86], ultimately hindering their involvement in the sustainable development plans of the NPs.

Unignorably, while community residents are the core stakeholders of NPs, their understanding of the impacts of NPs is often limited [27]. Even after losing the ability to extract resources from NPs, they still, naturally, strongly desire to maintain their traditional lifestyle [36,98]. In countries with limited resources for protection and conservation areas, illegal activities can become a particularly serious problem [118]. This issue has been widely reported in the management of NPs in developing countries in Africa, Asia, and South America [16]. It is crucial to address the impact of NP establishment on those who lived in and relied on natural environments like forests for their livelihoods when these activities were legal. To involve these residents in the management of NPs, they must perceive such actions as beneficial [119]. To this effect, finding alternatives to their original livelihoods becomes urgent. One approach is to encourage active participation in NP management through franchise operations, ecological management, or ecotourism, which can enhance their sense of belonging and ownership towards the NPs.

The tourism industry of the NP has the capacity to offer supplementary income, enhance the earnings of community residents, prevent the disruption of local environmental resources, contribute to the restoration of ecosystems and the environment, and gain broader community support. All of these aspects hold profound significance for the NP [25,89]. However, the tourism industry can also potentially bring about certain negative impacts on both the NP and its surrounding communities. Faced with the mounting pressure of increasing tourist demands, governments often invest significant resources into enhancing the infrastructure of the NP and its adjacent communities, such as roads, bridges, water supply, and power facilities [120]. But these developments can exert pressure on local natural resources and trigger a range of environmental issues, such as land degradation, water source contamination, fragmentation of landscapes, and a decline in biodiversity. Once tourism activities exceed the carrying capacity of NPs, they may lead to irreversible environmental damage [121]. Hu et al., 2022 [90], and Wang et al. [122] suggest that the rapid growth of the tourism industry can sometimes result in ecological, economic, and social degradation of local communities, especially pronounced in developing countries.

Therefore, to prevent the excessive depletion of natural resources by community residents and the unregulated development of the tourism industry, NP managers need to collaborate and strike a balance in assessing the impact of human activities on the natural environment of the NP. Achieving this objective requires concerted efforts among park managers, local communities, practitioners, tourists, and other stakeholders to explore the optimal approach for achieving a harmonious blend of NP development and environmental preservation.

### 4.8. Sources of the Articles behind National Park Community Management Research

Understanding and managing NP communities requires interdisciplinary research, encompassing fields such as ecology, environmental science, forestry, geography, sociology, and tourism studies. An increasing number of researchers are delving into issues related to NP communities from various perspectives and using various methodologies. Only through comprehensive approaches can we fully comprehend the complexity of NP communities and effectively promote their protection and sustainable development.

Analyzing the origin countries of scholarly publications and their global distribution characteristics (Figure 4) revealed a close relationship between the development process of NPs and their research background, which aligns with our expectations. The majority of research articles are concentrated in countries that established NPs early on, such as the USA, which has far more research publications than other countries. Following closely are other developed countries like the United Kingdom (206), Australia (204), and Canada (134). In developing countries, such as South Africa, China, Nepal, and India, where NPs were established later, the pace of research development is incredibly rapid. For instance, there has been a surge in research activity in China since the proposition of establishing an NP system for the first time in 2013 [123], especially after the formal establishment of the first batch of NPs in 2021. China has ambitious plans to establish about 50 NPs by 2035,

constructing the world's largest NP system [124], which has further stimulated research interest in this field.

## 5. Conclusions

We conducted a comprehensive, quantitative review of scientific literature on global NP community management spanning the past 33 years to determine the main research topics, trends, and publication patterns in this field. Our analysis revealed that research on NP community management is multifaceted, with a predominant focus on conservation. Key themes are reflected in the prevalence of keywords such as "Natural Resources", "Sustainable Development", "Stakeholders", "Community Management Models", "Tourism Development", and "Livelihoods". We constructed an interactive network diagram to illustrate the relationships among these themes.

The evolution of NP community management can be classified into three distinct phases: A slow development phase (Phase I, 1989–2000), a stable development phase (Phase II, 2001–2016), and a rapid growth phase (Phase III, 2017-present), indicative of an increasingly improved global NP management system. However, the distribution of research exhibits certain imbalances. Many studies originate from developed countries with well-established NPs, significantly outnumbering studies from developing countries. Among the top 20 countries with the most published research, 12 are developed countries, collectively contributing up to 72.18% of the total published research.

Furthermore, throughout the 33-year research period we examined in this study, there has been a noticeable shift in the focus of research. Initially, there was a singular emphasis on conserving natural resources; this has evolved to encompass a broader concern for human and societal well-being. To effectively align with the diverse developmental stages of global NP community management, it is imperative to foster interdisciplinary, cross-border, and cross-regional collaborations. This is particularly crucial for regions where NP management systems are in their infancy or relatively underdeveloped. By actively learning from countries with extensive management experience, these regions can accelerate their progress. The primary objective of new research should be seeking a balance and achieving synergy between the objectives of conserving NPs and advancing societal development. Ultimately, achieving the global sustainable development goals of NPs requires collective efforts from all stakeholders.

This paper provides a comprehensive overview of the NP community management research field based on bibliometric analysis. Our analysis outcomes are primarily based on data extracted from the WOS core database. Other databases such as Scopus, Dimensions, and PubMed are viable for bibliometric analysis. However, we opted to select a singular database to maintain data consistency and quality for our VOSviewer analysis.

It is important to note that this work differs from traditional bibliometric analyses. Instead of solely focusing on bibliometric metrics, our primary goal is to explore prominent topics within the research themes, their interconnections, and distinctive characteristics. By adopting this approach, we aim to gain a deeper understanding of the dynamics and trends in NP community management research over the past 33 years.

**Author Contributions:** Conceptualization, Y.Z., P.W. and G.W.; methodology, Z.W., A.S. and X.Z.; writing—original draft preparation, Y.Z.; writing—review and editing, A.S. and M.T.; visualization, Y.Z., Z.W. and X.Z.; funding acquisition, Y.Z., P.W. and G.W. All authors were committed to improving this paper and are responsible for the viewpoints presented in this work. All authors have read and agreed to the published version of the manuscript.

**Funding:** This research was supported by the China Scholarship Council (grant No. 202206760062), the Key R&D Program of Hubei Province, China (grant No. 2020BCA081), and the Fundamental Research Funds for the Central Universities (grant No. 2662020YLPY019). and UBC-APFNet National Park Research (grant No. GR025939).

**Data Availability Statement:** Data is available on request due to restrictions.

**Acknowledgments:** The author wishes to thank the anonymous reviewers who provided constructive comments and critical insight on this article.

**Conflicts of Interest:** The authors declare no conflict of interest.

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
