# Peer review of "Exploring the Main Determinants of National Park Community Management: Evidence from Bibliometric Analysis"

_forests, doi:10.3390/f14091850_

Round 1

Reviewer 1 Report

I read the article with great pleasure. The theme is very original. It fits perfectly with the profile of the magazine. It is written clearly, in accordance with the art of writing review articles. The management of national parks is an important aspect of their development. The authors also wonderfully refer to the issue of including stakeholders, including the community, in the decision-making process, which will certainly make management decisions effective. I am not aware of publications in the field of management that refer to this problem, hence the research gap is very valuable. It can certainly be interesting for foresters and institutions supervising national parks. It can also be an interesting read for the inhabitants of forest areas. 117 publications were used in the article. They are current and related to the topic. I have no objections. I do not report the need for additions. The discussion and conclusions are properly developed. The article contains numerous illustrations. They are very carefully crafted and add value to the consideration.

Author Response

Dear Reviewer:

     Thank you very much for your comments and suggestions, we have revised the manuscript accordingly. Our responses to each of your comments are listed below.

     In the following text, your comments are in black, our responses to the comments are in blue.

COMMENTS FOR THE AUTHOR:

Reviewer comments: I read the article with great pleasure. The theme is very original. It fits perfectly with the profile of the magazine. It is written clearly, in accordance with the art of writing review articles. The management of national parks is an important aspect of their development. The authors also wonderfully refer to the issue of including stakeholders, including the community, in the decision-making process, which will certainly make management decisions effective. I am not aware of publications in the field of management that refer to this problem, hence the research gap is very valuable. It can certainly be interesting for foresters and institutions supervising national parks. It can also be an interesting read for the inhabitants of forest areas. 117 publications were used in the article. They are current and related to the topic. I have no objections. I do not report the need for additions. The discussion and conclusions are properly developed. The article contains numerous illustrations. They are very carefully crafted and add value to the consideration.

Response

Thank you very much for carefully reading the article and providing valuable feedback. Your affirmation and encouragement have brought me great joy. Your evaluation of the discussion and conclusion, as well as your recognition of the illustrations, inspire me to delve deeper. Based on suggestions from other experts, I have made further revisions to the article, aiming to continuously enhance its quality. Once again, I sincerely appreciate your support.

Thanks and regards! 

Prof. Pengchang Wang

on behalf all co-authors

Reviewer 2 Report

The manuscript is well organized, having a robust methodological analysis and conveying the key aspects of the bibliometric analysis in a satisfactory manner.

1) It is kindly recommended authors to change the informal expression of “Hot Topics” from the manuscript title, replacing it with more formal expressions like: “Investigating the Key Aspects…….”, “Exploring the Main Determinants……”, “Analyzing the Parameters Affecting……..”

2) In the Abstract section authors are recommended to provide the exact period covering the documents applied to the conducted bibliometric analysis on the specific literature search databases, as well as the typical keywords or expressions under which the analysis took place.

3) Authors’ research objectives are to “reveal the mutual influence mechanisms of community management in the development process of NPs, thus positively influencing the sustainable development of NPs globally and human welfare.  However these research objectives are rather abstracted and general, thus, authors are recommended to specify -from a wider spectrum of consequences- only these statements and findings that have been derived from the specifically conducted bibliometric analysis.

4) The associations developed between the national parks (NPs) and the local communities can be introduced in the section of 1. Introduction. For example the exact roles of local citizens to use the sources of NPs should be conveyed: aesthetic, tourist guides, bus’s/car’s drivers for tour visitors, trading of anthems/products/gifts of local traditional-handmade production, renting rooms, other?, they should be succinctly discussed. Besides, the determinants of NPs evolution under: a) historic time back to 2-3 decades ago and b) themes overview regarding the key-aspects that yielded from the conducted bibliometric analysis. In this context it is also advisable authors to develop 3 separated subsections and succinctly present the distinct characteristics per each one of the Phases I, II, III, mentioned in the narrative flow.   

5) The main constraint of using the wide spectrum of 100+ citations is that almost all of them are covered only at a theoretical and descriptive manner, but this type of analysis is almost deprived from numerical data or quantitative information through which useful data could be also included in the analysis. Therefore, it is kindly proposed authors to revisit and retrieve from the same citations that are presented in each one of the 7 subsections 4.1 up to 4.7 and to develop 7 corresponding Tables, each one per these 7 subsections of main section 4, containing all numerical data or quantitative information that could be collectively provided in the form of these Tables.

6) The environmental contribution and concerns have been dispersed throughout the seven subsections of main section 4. Therefore it is recommended authors to gather and collectively represent these environmental concerns and challenges under one new and separated subsection 4.8, having also transferred all environment-related information to this new and autonomous section.

7) For literature completeness the following missing citations can be considered and cited at the revised manuscript: doi: 10.1007/s10668-022-02435-y

Author Response

Dear Reviewer:

Thank you very much for your comments and suggestions, we have revised the manuscript accordingly. Our responses to each of your comments are listed below.

In the following text, your comments are in black, our responses to the comments are in blue, and changes made to the manuscript are in red.

COMMENTS FOR THE AUTHOR:

Reviewer comments: The manuscript is well organized, having a robust methodological analysis and conveying the key aspects of the bibliometric analysis in a satisfactory manner.

1) It is kindly recommended authors to change the informal expression of “Hot Topics” from the manuscript title, replacing it with more formal expressions like: “Investigating the Key Aspects…….”, “Exploring the Main Determinants……”, “Analyzing the Parameters Affecting….”

Response

     Thanks for the suggestions. We modified the title: Exploring the Main Determinants of National Park Community Management: Evidence from Bibliometric Analysis.

2) In the Abstract section authors are recommended to provide the exact period covering the documents applied to the conducted bibliometric analysis on the specific literature search databases, as well as the typical keywords or expressions under which the analysis took place.

3) Authors’ research objectives are to “reveal the mutual influence mechanisms of community management in the development process of NPs, thus positively influencing the sustainable development of NPs globally and human welfare”.  However these research objectives are rather abstracted and general, thus, authors are recommended to specify -from a wider sectrum of consequences- only these statements and findings that have been derived from the specifically conducted bibliometric analysis.

Response

   In accordance with your valuable suggestions, adjustments have been made to the abstract section. We have incorporated a more detailed description of the literature retrieval timeframe (lines 26-27). Furthermore, the latter part of the abstract has been expanded upon to provide a more specific portrayal of our research focus (lines 28-36).

   Based on the literature from 1989 to 2022, utilizing 2,156 research papers from the Web of Science Core Collection database as the data source, visualizations were conducted using the VOSviewer software. Based on the results of network co-occurrence analysis, the initial focus of this field was on aspects of re-source conservation. However, with the convergence of interdisciplinary approaches, attention gradually shifted towards human societal well-being, emphasizing the “social-ecological” system. Furthermore, the current research hotspots in this field mainly revolve around issues such as “natural resources, sustainable development, stakeholder involvement, community management, sustainable tourism, and residents' livelihoods”. Effectively addressing the interplay of interests among these research hotspot issues has be-come an urgent topic for current and future research efforts. This exploration necessitates finding an ap-propriate balance between environmental conservation, economic development, and human welfare to promote the realization of long-term goals for sustainable development in NPs.

4) The associations developed between the national parks (NPs) and the local communities can be introduced in the section of 1. Introduction. For example the exact roles of local citizens to use the sources of NPs should be conveyed: aesthetic, tourist guides, bus’s/car’s drivers for tour visitors, trading of anthems/products/gifts of local traditional-handmade production, renting rooms, other?, they should be succinctly discussed. Besides, the determinants of NPs evolution under: a) historic time back to 2-3 decades ago and b) themes overview regarding the key-aspects that yielded from the conducted bibliometric analysis. In this context it is also advisable authors to develop 3 separated subsections and succinctly present the distinct characteristics per each one of the Phases I, II, III, mentioned in the narrative flow.  

Response

     Thank you very much for your suggestions. They have provided us with a highly valuable direction and insight. In this round of revisions, we have delved into the topic of “establishing connections among local communities” as you mentioned, in the "Introduction" section. This has been done in a more comprehensive manner to align with your views (Lines 88-104).

     Regarding the content of the section “the determinants of NPs evolution”, I sincerely appreciate the suggestions you've provided. Your insights are highly valuable to me. However, the primary focus of this article is inclined towards discussing various research hot topics related to the development of national park communities. In previous efforts, the categorization and summarization of the 2156 articles were not carried out based on the distinctive features of each stage within the first, second, and third phases. The focus was placed primarily on the exploration of the research themes of each individual article. Such an approach might have led to the oversight of temporal details.

     Hence, as things stand, it may be temporarily challenging for me to pursue the discussion along the lines you've proposed. Naturally, I wholeheartedly acknowledge the value of your suggestion. If deemed necessary, I intend to revise the article to incorporate this aspect when I have more ample time in the future. I sincerely appreciate your understanding and support.

5) The main constraint of using the wide spectrum of 100+ citations is that almost all of them are covered only at a theoretical and descriptive manner, but this type of analysis is almost deprived from numerical data or quantitative information through which useful data could be also included in the analysis. Therefore, it is kindly proposed authors to revisit and retrieve from the same citations that are presented in each one of the 7 subsections 4.1 up to 4.7 and to develop 7 corresponding Tables, each one per these 7 subsections of main section 4, containing all numerical data or quantitative information that could be collectively provided in the form of these Tables.

Response

     Your suggestions are of great value to me. Towards the conclusion of the article, I put forth a viewpoint that “It is important to note that this work differs from traditional bibliometric analyses. Instead of solely focusing on bibliometric metrics, our primary goal is to explore prominent topics within the research themes, their interconnections, and distinctive characteristics. By adopting this approach, we aim to gain a deeper understanding of the dynamics and trends in NP community management research over the past 33 years”. (Lines 706-710).

      With the existence of this viewpoint, on one hand, it helps steer away from the approach commonly found in similar “quantitative” articles. On the other hand, it is true that we did not undertake the collection and compilation of quantitative information in our intended writing process. As you rightly suggested, we concur that adding tables would indeed introduce a “qualitative & quantitative” dimension to the article. However, it cannot be overlooked that the inclusion of this section would result in a lengthier article. Therefore, during the data collection phase, we directed more of our efforts toward the discussion of “qualitative” aspects. I am truly appreciative of your attention to the article and your valuable suggestions.

6) The environmental contribution and concerns have been dispersed throughout the seven subsections of main section 4. Therefore it is recommended authors to gather and collectively represent these environmental concerns and challenges under one new and separated subsection 4.8, having also transferred all environment-related information to this new and autonomous section.

Response

    In accordance with your suggestions, we have reorganized the content from the original sections 4.1-4.6 concerning environmental issues and challenges. These have been consolidated and placed within a new section titled “4.7 Environmental Issues Challenges Faced by National Park Communities”. Concurrently, the previous section 4.7 has been renumbered as 4.8 (Lines 592-651). Your guidance is greatly appreciated.

7) For literature completeness the following missing citations can be considered and cited at the revised manuscript: doi: 10.1007/s10668-022-02435-y

Response

     Thank you for your suggestion. Based on the literature, we have incorporated new citations (Lines 579-582) and referenced source [108] in the bibliography.

     Furthermore, urbanization and agricultural expansion are considered the two major driving factors leading to deforestation. The logging of forests results in a gradual degradation of forest services and has both direct and indirect impacts on rural and urban societies

Thanks and regards! 

Prof. Pengchang Wang

on behalf all co-authors

Reviewer 3 Report

This paper provides a bibliometric analysis of the scientific literature concerning community management of National parks on a global scale. There are several issues which need to be addressed.

In section 2.1 there is a mention regarding a focus group to identify synonyms or near-synonyms of the term “Community”. The authors should provide more details:

- who were the participants to this focus group?

-how were they selected?

-what questions/themes were included in the interview guide?

-when and how the focus group was organized?

In section 4.5 "Tourism development", authors should also mention the challenges in managing tourism in NP identified in the analyzed papers.

In the final part of the paper, the limitations of the study should be highlighted as well as future research directions.

Author Response

Dear Reviewer:

Thank you very much for your comments and suggestions, we have revised the manuscript accordingly. Our responses to each of your comments are listed below.

In the following text, your comments are in black, our responses to the comments are in blue, and changes made to the manuscript are in red.

COMMENTS FOR THE AUTHOR:
Reviewer comments: This paper provides a bibliometric analysis of the scientific literature concerning community management of National parks on a global scale. There are several issues which need to be addressed.

  1. In section 2.1 there is a mention regarding a focus group to identify synonyms or near-synonyms of the term “Community”. The authors should provide more details:

   -who were the participants to this focus group? -how were they selected?

   -what questions/themes were included in the interview guide?

   -when and how the focus group was organized?

Response

     Thank you for your suggestion, your concerns and inquiries are exactly what we were worried about when writing the article. Indeed, the accuracy of the term “Community” or rather the correctness of choosing it, determines the feasibility of the article. Therefore, we explored the theme of this article by organizing a "focus group" and conducting literature screening experiments.

     Our focus group is primarily composed of research institutions from two countries (Huazhong Agricultural University, China; University of British Columbia, Canada). The team is mainly comprised of professors and master's students, with a significant presence from personnel at the National Park Research Center of the University of British Columbia (https://nationalparks.forestry.ubc.ca/) and the Asian Forest Research Center (https://afrc.forestry.ubc.ca/people/).

   During the period of February to March 2023, a total of three discussions were held through both online and offline meetings. The main topic of these discussions revolved around synonyms or similar terms for the "community" within the context of national parks. Additionally, we also referenced the results of previous community research conducted in Shennongjia National Park in China and Banff National Park in Canada (Fig 1). Taking into account the perspectives of local residents and government personnel on the understanding of the "Community" within the context of national parks, we subsequently provided a summary of our exploratory findings regarding the community, as presented in lines 130-132 of the article.

Data collection:

    The keyword strings used were TS = (“National Park*”) AND TS= (“Community Management*” OR “Community-Based Management*” OR “Community Conservation*” OR “Local People*” OR “Buffer Zone*” OR Co-management* OR Indigenous*).

  1. In section 4.5 "Tourism development", authors should also mention the challenges in managing tourism in NP identified in the analyzed papers.

Response

     Thanks for the suggestions. We have revised section 4.5 according to the recommendations and included additional content regarding “Challenges in the management of national park tourism”. The updated version can be found in lines 542-549 of the article.

     Simultaneously, it should not be overlooked that the vigorous growth of the tourism industry has brought about certain challenges for the sustainable development of NPs. The substantial tourism activities around NPs unavoidably inflict a certain degree of damage and disruption upon the natural systems, concurrently leading to shifts in the so-cio-cultural environment [15]. Nevertheless, by actively guiding community engagement and implementing effective management measures, NP managers can minimize these disturbances, thereby achieving the dual objectives of ecological conservation and sustainable development for the NP [96,97].

  1. In the final part of the paper, the limitations of the study should be highlighted as well as future research directions.

Response

Thanks for the suggestions. The main limitations of this article are primarily the selection of databases and software, details can be found in lines 701-705.

   Future research directions are mainly focused on: The primary objective of new research should be seeking a balance and achieving synergy between the objectives of conserving NPs and advancing societal development. (lines 697-700)

Thanks and regards! 

Prof. Pengchang Wang

on behalf all co-authors

Round 2

Reviewer 2 Report

Authors revised their manuscript in a systematic and thorough manner, thus, it can be accepted for publication at the Forests journal as is.